# Limitations of Tamoxifen Application for In Vivo Genome Editing Using Cre/ER^T2^ System

**DOI:** 10.3390/ijms232214077

**Published:** 2022-11-15

**Authors:** Leonid A. Ilchuk, Nina I. Stavskaya, Ekaterina A. Varlamova, Alvina I. Khamidullina, Victor V. Tatarskiy, Vladislav A. Mogila, Ksenia B. Kolbutova, Sergey A. Bogdan, Alexey M. Sheremetov, Alexandr N. Baulin, Irina A. Filatova, Yulia Yu. Silaeva, Maxim A. Filatov, Alexandra V. Bruter

**Affiliations:** 1Center for Precision Genome Editing and Genetic Technologies for Biomedicine, Institute of Gene Biology, Russian Academy of Sciences, 119334 Moscow, Russia; 2Institute of Gene Biology, Russian Academy of Sciences, 34/5 Vavilov Street, 119334 Moscow, Russia; 3Federal State Budgetary Institution “N.N. Blokhin National Medical Research Center of Oncology”, Ministry of Health of the Russian Federation, Kashirskoe Sh., 24, 115478 Moscow, Russia; 4Head Center of Hygiene and Epidemiology of Federal Medical Biological Agency of Russia, 1st Pekhotniy Pereulok, 6, 123182 Moscow, Russia

**Keywords:** Cre-LoxP, Cre-ER^T2^, tamoxifen, genetically engineered mice, genome-edited mice, conditional knockout, cyclin-dependent kinases 8/19

## Abstract

Inducible Cre-dependent systems are frequently used to produce both conditional knockouts and transgenic mice with regulated expression of the gene of interest. Induction can be achieved by doxycycline-dependent transcription of the wild type gene or OH-tamoxifen-dependent nuclear translocation of the chimeric Cre/ER^T2^ protein. However, both of these activation strategies have some limitations. We analyzed the efficiency of knockout in different tissues and found out that it correlates with the concentration of the hydroxytamoxifen and endoxifen—the active metabolites of tamoxifen—measured by LC-MS in these tissues. We also describe two cases of Cdk8^floxed/floxed^/Rosa-Cre-ER^T2^ mice tamoxifen-induced knockout limitations. In the first case, the standard scheme of tamoxifen administration does not lead to complete knockout formation in the brain or in the uterus. Tamoxifen metabolite measurements in multiple tissues were performed and it has been shown that low recombinase activity in the brain is due to the low levels of tamoxifen active metabolites. Increase of tamoxifen dosage (1.5 fold) and duration of activation (from 5 to 7 days) allowed us to significantly improve the knockout rate in the brain, but not in the uterus. In the second case, knockout induction during embryonic development was impossible due to the negative effect of tamoxifen on gestation. Although DNA editing in the embryos was achieved in some cases, the treatment led to different complications of the pregnancy in wild-type female mice. We propose to use doxycycline-induced Cre systems in such models.

## 1. Introduction

Many genes are critically important for different stages of embryonic development and therefore animals with their constitutive knockout die during embryonic development [1,2,3]. Conditional knockout organisms are required to study the functions of the affected genes on later stages of development and throughout the adult lifespan. One of such gene is *Cdk8* encoding cyclin-dependent kinase 8, knockout of which is lethal for mouse embryos at the preimplantation stage [4]. Despite being a cyclin-dependent kinase, CDK8 does not take a direct part in cell cycle regulation. It represents the main part of the mediator complex kinase module, which regulates Pol II activity, besides having a number of phosphorylation targets incorporated in various signaling pathways [5]. CDK8 has a paralog, named CDK19, which presumably substitutes it in many aspects, and this is why the *Cdk8* knockout phenotype does not typically emerge in adult mice [4,6]. However, CDK19 cannot replace CDK8 in embryonic development, and the constitutive knockout of *Cdk8* leads to impairments in development [4,7]. 

Producing a conditional knockout animal requires a robust, reliable and effective knockout activation system with minimal side effects. Cre-ER^T2^ fusion protein, commonly used as an inducible activator, consists of a Cre-recombinase domain, and a mutant estrogen receptor (ER^T2^) domain which binds 4-hydroxytamoxifen (4-OHT) [8,9], or similar metabolites such as endoxifen [10], but not the endogenous estrogens [8,9,11]. After binding, the fusion protein is translocated into the nucleus. Tamoxifen (TAM) could be used to induce gene recombination in vivo [12], since it is converted into 4-OHT in the liver [13,14].

Although this system is widely used, it has still been poorly described in many aspects and has a number of drawbacks. For example, in many cases tamoxifen-induced gene editing demonstrates low efficiency [15]. Tamoxifen by itself also has side effects on naturally occurring processes in organisms that are poorly investigated; for example, it has been shown that its application, intended to activate knockout of the androgen receptor gene, alters the hormone profile per se, most notably, that of testosterone [16]. Moreover, teratogenic effects of tamoxifen were described in clinical practice even in cases when tamoxifen treatment was performed before the pregnancy; tamoxifen administration may also lead to miscarriage [17,18].

In the current work we compare two different protocols of tamoxifen administration (oral gavage and intraperitoneal injection) by assessing knockout effectiveness and tamoxifen tissue distribution.

Moreover, we describe two cases when obtaining knockouts in Cdk8^floxed/floxed^/Rosa-Cre-ER^T2^ mice was complicated. In the first case, the standard scheme of tamoxifen administration had not led to knockout formation in the brain due to the limited penetration of tamoxifen and its metabolites into the brain, which was confirmed by HPLC-MS. This problem can partially be solved by increasing the dose of administered tamoxifen. However, low knockout efficiency in the uterus was not related to the tamoxifen distribution.

In the second case, knockout induction at the late stages of embryonic development led to the negative effect of tamoxifen on gestation. Although DNA editing occurred in several embryos, tamoxifen led to miscarriages and stillbirths in most observed cases.

## 2. Results

### 2.1. Knockout Efficiency and Different Tamoxifen Administration Routes

Tamoxifen is mainly administered by two routes: oral gavage or intraperitoneal injections. First, we compared feeding (3 mg per day) and injection methods (2 mg per day) once a day for 5 consecutive days. These schemes of tamoxifen administration are widely used for knockout activation [19,20,21,22].

We demonstrated that the tamoxifen feeding method is appropriate for knockout of the target gene in mice organs such as lungs, liver, and spleen (Figure 1A). However, we observed only moderate CDK8 decrease in the uterus and did not observe the CDK8 decrease in the brain of the knockout mice. No CDK8 protein was detected in kidneys even of the wild type animals.

We also administered tamoxifen by intraperitoneal injection in a volume of 100 µL (2 mg) for 5 consecutive days. Likewise, this method is suitable for producing complete knockout in lungs, liver, spleen, and is less effective in the brain of the tamoxifen-injected mouse (see Figure 1B). However, the knockout efficiency in the brain was higher in the case of injection than in the case of feeding.

To conclude, both feeding and injection administration are appropriate for knockout of *Cdk8* in several organs of the tamoxifen inducible-driven mice. However, both methods failed to produce complete knockout in the brains of the double transgenic mice.

Because the knockout efficiency increased (see Figure 1A,B) after the change of the administration route, we presupposed that it may be further enhanced with the augmentation of the tamoxifen dosage.

### 2.2. Enhanced Scheme of Tamoxifen Injection Activates a Total Knockout of Target Gene

As shown above, the standard activation scheme did not provide a sufficient knockout in the brain and in the uterus. It is noteworthy that the excision of the 2nd exon of *Cdk8* in the brain after a 5-day activation scheme had not been demonstrated in the initial paper describing the development of the *Cdk8* conditional knockout mice [6] which argues that CDK8 is disposable in adult mice. At the same time, the brain is one of the most important organs for maintaining vital activity and homeostasis of the organism and it has been shown that CDK8 is expressed in the brain [23,24], although its role remains unclear. Therefore, we tried to achieve a complete *Cdk8* knockout in the brain by increasing the single dose of tamoxifen by 1.5 fold and the duration of injections from 5 up to 7 days, which constitutes 21 mg of tamoxifen per mouse (3 mg per day). As a result, we substantially increased the *Cdk8* knockout rate in the brain with the advanced injection scheme (see Figure 1C). However, although knockout efficiency slightly increased in the uterus after applying the improved scheme, we still could not get a complete *Cdk8* knockout.

Thus, the enhanced scheme of injection improves *Cdk8* knockout in the mouse brain but not in the uterus.

As PCR is a more sensitive method than Western blotting, to check if unedited DNA remained, we performed genomic PCRs using as template genomic DNA extracted from tamoxifen-treated homozygous mice organs. Twelve organs, namely heart, ovary, uterus, liver, lung, brain, skeletal muscle, kidney, small intestine, thymus, spleen and bone marrow were analyzed.

As predicted, we observed incomplete excision in ovary and uterus, which affirms that Cre/ER^T2^ system is unsuitable for studying knockouts in the female reproductive system.

To our disappointment, the knockout in the brain was found to be still incomplete, which indicates that the Cre/ER^T2^ system must be used very carefully when knockout in the brain must be achieved.

We also detected some floxed exon 2 remains in the small intestine (Figure 2B, samples of only one treated mouse are shown). In all other organs, the excision was successful.

### 2.3. LC-MS/MS Reveals Distribution Variability and Higher Injection Efficiency

To clarify the causes of insufficient *Cdk8* knockout activation in the brain and uterus, concentrations of tamoxifen and its main active metabolites, namely endoxifen and 4-hydroxytamoxifen, were measured in liver, uterus, lungs and brain. To deduce the short-term influence of the administration method, tamoxifen kinetics in mice were analyzed at two time points: 4 and 17 h after drug administration.

The distribution was assessed in 4 groups of 3 adult C57Bl6/J female mice (mean mass 19 g). Subjects were grouped by route of administration and time points. Each mouse was treated with a single 3 mg dose of tamoxifen. Its metabolites were extracted from homogenized organs with methanol and run through Solid Phase Extraction (SPE) cartridges. The extracts were analyzed by LC-MS/MS. SPE recovery rate was approx. 87%.

Notably, average liver tamoxifen concentration constituted 4.1 × 10^5^ ng/g at 4 h when injected (ranging (1.86–7.00) × 10^5^ ng/g), which, taking liver mass (m = 1.67 g [25]) into account, makes up 0.684 mg (22.8%) of total 3 mg applied (Figure 3).

We discovered that the administration method plays an important role in drug absorption: fed mice showed substantially lower metabolite concentrations than those that were injected for each organ and time point. Although average injected tamoxifen absolute levels in uteri constituted 4.9 × 10^5^ and 2.2 × 10^5^ ng/g, at 4 and 17 h, respectively (Figure 4A, Appendix A), an extreme difference in metabolite concentration was found in it with respect to the route of administration: ratios of uterine group means between injection and feeding method samples reached two orders of magnitude, ranging from 12.8 to 230.8 (Table 1, Figure 5). Endoxifen and OH-Tamoxifen ratios are distributed more uniformly around a 10-fold difference (median) for organs other than the uterus (Table 1, Figure 5). However, as was shown above, intraperitoneal delivery had not led to complete knockout in the uterus.

Despite the lowest concentration of tamoxifen itself, the highest metabolite concentration at both time points regardless of administration route was detected in the lung (Figure 4A, Appendix A). We also found that among all organs, brains typically accumulated the smallest overall amount of metabolites and tamoxifen (0.18 × 10^3^ and 6.64 × 10^3^ ng/g of endoxifen and OH-tamoxifen, respectively) which correlates well with observed knockout efficiency measured by WB. Contrarily, WB shows ineffective activation in the uterus, although metabolite concentrations in the liver and uterus proved to be comparable after 17 h.

### 2.4. Effects on Pregnancy

Knockout induction during different stages of pregnancy could be a useful tool to investigate the role of certain genes in post-implantational stages of embryonic development. Recently it has been shown that application of tamoxifen is suitable to activate at least in some cells the expression of the reporter gene initially silenced by the floxed STOP-cassette [22]. In this work, 12 mg of the tamoxifen administered with chow caused fetal death and decrease of the tamoxifen dosage 2–4 fold allowed to obtain pups with at least partially activated transgene. In other work, it has been shown that tamoxifen administration during pregnancy allows viable offspring; however, a teratogenic effect has been observed. Moreover, doses about 1 mg per mouse were used in this work [26], which is probably insufficient to obtain knockouts.

To investigate whether tamoxifen is suitable not only for the activation of the genes, but also to achieve complete knockouts, we assessed 2nd exon excision effectiveness in Cdk8^fl/wt/^R26-Cre-ER^T2 +/−^ embryos starting from E10.5 (E—days of embryo development). A Cdk8^fl/fl/^ROSA-Cre-ER^T2 +/+^ male animal of C56Bl/6J background was crossed with a wild-type female mouse (C57Bl/6J) to obtain Cdk8^fl/wt^/ROSA-Cre-ER^T2 +/−^ embryos. *Cdk8* heterozygosity does not manifest phenotypically in our experience (as has also been shown in the work of [4]). Upon vaginal plug observation, the female mouse was injected with 1.5 mg tamoxifen doses for 3 days starting at 10.5 dpc (days post coitum). As a result, premature birth at 18.5 dpc occurred, resulting in 3 dead pups and 2 nonviable pups that died during the next 3 days. Though excision of the floxed sequence in all the pups was confirmed by PCR (see Figure 6A), this system cannot be used for knockouts production at embryonic stages as is. An identical experiment starting at 17.5 dpc resulted in weakness of labor and after caesarean section at 20.5 dpc, 7 nonviable pups were obtained. Moreover, excision analysis demonstrated incomplete knockout in all embryos (Figure 6B).

To further investigate the applicability of tamoxifen administration during pregnancy and to prove that effects described above are not due to partial *Cdk8* knockout, we decided to test the tamoxifen action on wildtype pregnant mice. We crossed hybrid females F_1_ CBA × C57Bl/6 with hybrid males with the same genetic background. An experiment was conducted involving three pregnant mice, and the results are outlined in Table 2. After observing the copulatory plug, these mice received tamoxifen orally in the dosage 3 mg of tamoxifen per day during 4 days. Mouse #1 started to receive tamoxifen at 15.5 dpc. After administration of two doses of tamoxifen at the 17.5 dpc, this animal went into premature labor which led to the death of pups and mother. Then, we decided to decrease tamoxifen dosage 2-fold (1.5 mg). Mouse #2 also started to receive tamoxifen at 15.5 dpc (1.5 mg of tamoxifen per day during 5 days). Labor occurred at 21 days of pregnancy and progressed pathologically; pups were stuck in the birth canal and eventually the mother and pups died. Mouse #3 started to receive tamoxifen at 10.5 dpc (1.5 mg of tamoxifen per day during 5 days), premature labor occurred at 17.5 dpc, and the pups died. Having taken the presented preliminary results into consideration, the experiment involving CBA × C57Bl/6 hybrid mice was stopped at the demand of the IGB RAS bioethical commission. A suggestion was made that such results were observed due to using hybrid mice, since most publications describe experiments carried out on the pure C57Bl/6 strain. Mice #4–6 received 1.5 mg of tamoxifen per day during 3 days from 10.5 dpc. Premature labor occurred at 18.5 dpc, and the pups died. Mice #7–8 received 1.5 mg of tamoxifen per day during 3 days from 15.5 dpc with the same effect. This experiment was also stopped at the demand of the IGB RAS bioethical commission.

## 3. Discussion

Tamoxifen is ubiquitously used for conditional gene editing, but often without proper controls and verifications. Moreover, its off-target effects, and correlation between different protocols of its administration and tissue distribution of tamoxifen and its active metabolites (OH-tamoxifen and endoxifen) are poorly investigated. Robinson and colleagues [27] measured concentrations of tamoxifen, OH-tamoxifen and endoxifen in the blood samples of rats and mice after oral administration of tamoxifen. However, the efficiency of nuclear translocation of Cre-ER^T2^ depends on the intracellular concentration of tamoxifen’s active metabolites, and therefore their concentration in the tissue determines the efficiency of gene editing. Another group [28] measured concentrations of tamoxifen and its active metabolites in the murine brain. 

In the current article we have shown that the concentration of tamoxifen and its active metabolites is significantly reduced in the brain in comparison to other organs (such as spleen, liver and lungs). This leads to poor knockout efficiency in the brain, which is increased to the absence of detectable protein, if the dosage of administered tamoxifen is also increased. However, we could not achieve full knockout in the brain in our experiments, on the genomic level. Gene switch-off in the brain may lead to the different phenotypic manifestations such as changes in behavior and cognitive functions, or somatic disorders caused by impaired function of the hypothalamic-pituitary axis. Therefore, when the phenotype of conditional knockout is investigated, it is important to prove that knockout production was achieved in different tissues, including the brain. One of the possible reasons why knockout production in the brain induced by tamoxifen administration is impaired is that penetration of tamoxifen and its active metabolites through the brain-blood barrier may be reduced. Similar effects were demonstrated for the hematotesticular barrier by Patel et al. [29]. However, in the work [28] it is stated that disruption of the blood-brain barrier by occlusion of the middle cerebral artery causes no changes in the ipsilateral hemisphere concentration of any tamoxifen metabolites.

Another tissue where the conditional knockouts are not as easily achieved is the uterus and the ovaries. In the present article we show that despite relatively high concentration of tamoxifen and its metabolites, Cre-ER^T2^-induced knockout is not achieved in the uterus and in the ovaries. The knockout was not induced even when the concentration of tamoxifen was increased, or the route of administration was changed to intraperitoneal. The mechanism of this impaired activity should be investigated further and could be related to the tissue-specific activity of the promoter driving Cre-ER^T2^ expression or other factors, for example, presence of the wild type ER receptor in the cells.

Tamoxifen is an antagonist of estrogen receptors and may influence molecular signaling besides induction or inactivation of targeted genes. Conditional knockout models require stringent controls for off-target effects. It has been shown that, in male mice, tamoxifen treatment led to the decrease in serum levels of gonadotropins and testosterone levels, and expression of several genes in Leydig cells (StAR, Cyp17a1, Insl3), even after the end of tamoxifen treatment [16]. Moreover, it has been shown that single tamoxifen injection in juvenile mice led to adverse effects in the testis and in the reproductive endocrine system that persisted long-term [29]. 

Dramatic effects are also observed when tamoxifen administration is applied to induce activation or inactivation of embryonic genes in utero. Teratogenic effects of tamoxifen were described in clinical practice even in case when the tamoxifen treatment was performed before the pregnancy; moreover, tamoxifen administration may lead to miscarriage [17,18]. In the experiments performed on mice and rats, it has been shown that tamoxifen administration during pregnancy can lead to craniofacial malformations among the embryos [26,30], increase the probability of tumor formation [31], and disrupt the reproductive system development [32,33]. 

However, recent publications have shown that tamoxifen administration can lead to successful excision of floxed STOP-cassette and activation of β-galactosidase gene expression in utero [22]. Generally, emergence of the phenotype (if any) may require higher efficiency of floxed sequence excision in the case of the knockout activation than is needed in the case of expression activation, so we examined if decreased doses are enough to achieve knockouts. However, our present data shows adverse effects on pregnancy even for the lowest used doses and number of injections in all examined animals, including premature birth, stillbirth, and other pathologies in labor activity, confirming other published works in the field [34,35]. Overall, data presented in the current article suggests that Cre/Lox tamoxifen-induced models cannot be used for conditional knockouts in embryos. In this work we demonstrated total absence of floxed 2nd exon in pups treated with tamoxifen in utero at 10.5–12.5 dpc, but the pups, even if alive at birth, were not viable. In the pups treated with tamoxifen at 17.5–19.5 dpc, we observed only partial excision of exon 2, which is probably explained by increased size of the embryos. Taken together with the low knockout rate in the uterus and ovaries, it makes the recent Cre-ER^T2^ system completely inappropriate for studying knockout effects in the female reproductive system.

Probably, novel more effective variants of Cre-ER^T2^ recombinase should be designed to avoid hormonal side-effects of tamoxifen when it is used for activation or inactivation of genes. One such model uses chimeric Cre-ER^T2^ recombinase which under tamoxifen action not only performs editing of the target gene but also edits self Cre-ER^T2^ recombinase gene resulting in a constitutively active Cre [36]. That allows genome editing with lower tamoxifen doses. Another approach to induce genome editing avoiding hormonal effects of tamoxifen that seems to be preferable in this application is the doxycycline activation system.

Altogether, our present data cautions against interpreting lack of phenotype in Cre/Lox models as lack of effect of gene deletion, without confirmation of complete knockout in all critical tissues, especially in the brain and the uterus. On the other hand, we suggest caution on over-interpreting effects of such knockouts, without using thorough controls for effects of tamoxifen itself, especially in studying effects in embryos and in the reproductive system. Overall, despite major advances gained from using inducible tamoxifen Cre/Lox models, a new generation of activators of genetic constructs could not only increase the efficiency of inducible genetic modification, but also allow study of processes that are currently too much affected by limitations of the model itself. 

## 4. Materials and Methods

### 4.1. Animals

C57BL/6J mice and B6.129-Gt(ROSA)26Sortm1(cre/ERT2)Tyj/J (strain number 008463) hereinafter referred to as R26-Cre-ER^T2^, were purchased from the Jackson Lab. Cdk8^fl/fl^ mice were kindly provided by Genentech and initially described by the Firestain group [6]. Animals were maintained under controlled room conditions (22–24 °C and a 14 h light:10 h dark photoperiod). Mice were given ad libitum access to food and water. Approximately two months old mice were used in the experiments. All manipulations with animals were performed according to the Local Bioethical Committee recommendations and according to the Declaration of Helsinki (1996).

To achieve the pregnancies, female mice were crossed overnight with males. The date of pregnancy was determined by observation of the vaginal plug.

### 4.2. Tamoxifen Administration 

Tamoxifen was administered by two different routes: by intraperitoneal injection and by oral administration.

In the case of oral administration one tablet of tamoxifen citrate (Salutas Pharma, Barleben, Germany) was suspended in saline solution (0.9% NaCl) to reach the final concentration of tamoxifen (20 mg/mL). Feeding of animals was performed daily once a day during 5 days. Each animal received 150 μL of prepared tamoxifen suspension per day, resulting in total administration of 15 mg of tamoxifen per mouse. For LC-MS/MS analysis, mice received tamoxifen once at the dose 3 mg either orally or intraperitoneally. In the case of intraperitoneal injection, tamoxifen (Sigma-Aldrich, Burlington, MA, USA) powder was dissolved in corn oil (Sigma-Aldrich, Burlington, MA, USA) in the proportion of 20 mg of tamoxifen per 1 mL of corn oil. To achieve better dissolution of tamoxifen the solution was incubated at 37 °C for 4 h accompanied by constant shaking. Injection was performed intraperitoneally by insulin syringe. Initially, mice received injections of 100 μL of prepared solution daily for 5 days, resulting in a total administration of 10 mg of tamoxifen per mouse. After the discovery of the incomplete knockout in the brain, it was decided to increase the dosage up to 150 μL per day and total duration of 7 days, thus resulting in total administration of 21 mg of tamoxifen per mouse.

### 4.3. Genotyping and Detection of Excision at the DNA Level

Genotyping of the R26-Cre-ER^T2^ mice (B6.129Gt(ROSA)26Sortm1(cre/ERT2)Tyj/J) was performed according to Jackson’s lab protocol. Genotyping of the Cdk8^fl/fl^ mice was performed by PCR using primers Cdk8-F and Cdk8-R2 placed on the two sides of a LoxP (Figure 2A) site resulting in 208 band for a wild type allele and 242 band for a floxed allele (see Figure 7). Hot Start Taq DNA polymerase and Turbo buffer (Evrogen, Moscow, Russia) were used for genotyping. 1% agarose gel was used for Rosa-Cre-ER^T2^ whereas 2% agarose gel was used for Cdk8^fl/fl^.

Double transgenes were genotyped for both Cdk8^fl/fl^ and R26-Cre-ER^T2^ and homozygous for Cdk8^fl/fl^ and at least heterozygous for Rosa-Cre-ER^T2^ mice enrolled in the experiments. Exon 2 excision in distinct organs and heterozygous embryos was detected by PCR with primers Cdk8-F and Cdk8-R. Presence of edited allele resulted in a band about 414 bp. A single 208 bp band without 242 bp band in PCR with Cdk8-F and Cdk8-R2 confirmed the presence of wild type allele and complete excision of floxed sequence in transgenic allele in heterozygous embryos (Figure 2A), while no PCR product was detected after complete excision of exon 2 in the tissues of homozygous mice (Figure 2). Presence of both 208 bp and 242 bp bands indicated incomplete excision of the exon 2 (Figure 7). The sequences of used primers are given in Appendix A.

### 4.4. Immunoblotting

Organ samples obtained from transgenic and control mice were homogenized and ground in the presence of liquid nitrogen. Then samples were lysed for 30 min on ice in RIPA buffer containing 50 mM Tris-HCl pH8, 150 mM sodium chloride (NaCl), 0.1% sodium dodecyl sulfate (SDS), 1% Nonidet P-40 (NP-40), 2 mM PMSF (Helicon, Moscow, Russia) and protease inhibitor cocktail (PIC, Sigma-Aldrich, Burlington, MA, USA). Total protein concentration in lysates was determined by the Bradford method. Lysates were separated by SDS polyacrylamide gel electrophoresis (PAGE) (60 μg total protein per lane) and transferred onto a 0.2 μm nitrocellulose membrane (Bio-Rad, Hercules, CA, USA). Non-specific protein-antibody interactions were blocked with 5% skimmed milk for 30 min at room temperature. The membranes were blotted with primary anti CDK8 specific (sc-1521, SantaCruz, Santa Cruz, CA, USA) and anti β-actin specific antibodies (A2228, Sigma-Aldrich, Burlington, MA, USA) diluted in Tris buffered saline with Tween 20 (TBS-T; 1:1000) supplemented with 1% BSA overnight at 4 °C. Then, the membranes were washed with TBS-T and incubated with the secondary antibodies conjugated with horseradish peroxidase (Sigma-Aldrich) diluted in 5% skimmed milk in TBS-T (1:1000) for 1 h at room temperature and then washed with TBS-T. The luminescent detection of proteins was performed with the Clarity Western ECL Substrate (Bio-Rad) using iBright FL1500 Imaging System (Invitrogen, Carlsbad, CA, USA).

### 4.5. Liquid Chromatography with Tandem Mass Spectrometry

To determine the distribution of tamoxifen and its metabolites within a mouse body, we measured absolute concentrations of substances in several organs: brain, uterus, liver and lungs using liquid chromatography with tandem mass spectrometry (LC-MS/MS).

Twelve adult (2 months) wild type C57Bl/6J female mice had been divided into two groups of six animals, each administered with 3 mg dose of tamoxifen as previously described, using either intraperitoneal injection or oral gavage.

Each group had been subdivided into two subgroups according to the time passed from drug administration to sample gathering: either 4 or 17 h, which are thought to be time points of maximum serum concentration and elimination half-life, respectively [27].

#### 4.5.1. Reagents

Tamoxifen (Tam, 85256), Z-endoxifen (Endo, SML2368) and 4-hydroxytamoxifen (OH-Tam, 94873) standards were obtained from Sigma-Aldrich, Burlington, MA, USA. The Solid Phase Extraction columns (SPE), STRATA-X-C (3 mL, 30 mg), were obtained from Phenomenex, Torrance, CA, USA. 0.1% solution of formic acid was used as mobile phase A (A) and Acetonitrile as a mobile phase B (B).

#### 4.5.2. Sample Gathering and Preparation

Mice were sacrificed by cervical dislocation. Whole organs were excised within 5 min after death, gently rinsed with PBS (phosphate-buffered saline) and snap-frozen in liquid nitrogen.

All samples were homogenized with pestle and mortar in liquid nitrogen. Approximately 50 mg of powder was transferred into clean microcentrifuge tubes, covered with ice-cold methanol (1:6 *w*/*v*) [37], lightly vortexed and set on ice. Each sample was sonicated using QSONICA Q125 (Qsonica, Newtown, CT, USA) at 40% amp in two rounds of 2 s with a 10 s rest on ice. Homogenates were incubated for 5 min at 0 °C and then centrifuged for 5 min at 5000× *g* 1 °C, and liquid phases were transferred into clean prechilled tubes to obtain extracts. Frozen organs and extracts were stored at −80 °C.

Metabolite extract purification was performed by solid phase extraction using a previously described method [38]. Dried eluates were redissolved in 500 μL of 50:50 AcN:H_2_O and used for analysis.

#### 4.5.3. Sample Analysis

For concentration quantifications a Shimadzu LCMS-8060 tandem mass spectrometer (Kyoto, Japan) was used in positive electrospray ionization mode. Tamoxifen and its metabolites were separated by isocratic HPLC in a 0.4 mL/min flow of A:B 40:60 (*v*/*v*) on a Shim-pack GIST C_18_ 150 × 3.0 mm column (Shimadzu, Kyoto, Japan). Interface temperature and desolvation line were set at 250 °C; heat block temperature was 400 °C. The optimized operation conditions were determined for a mixture of analyte standards as presented in Table 3. Every measurement was run in two technical replicates.

## Figures and Tables

**Figure 1 ijms-23-14077-f001:**
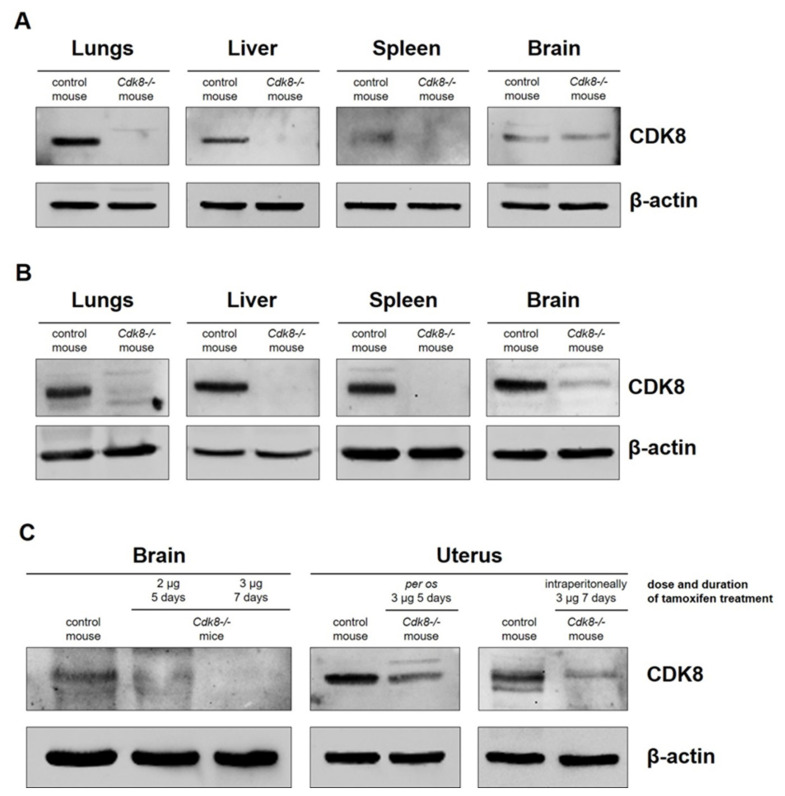
Detection of the tamoxifen-induced *Cdk8* knockout in different organs of mice: (**A**) feeding method of tamoxifen administration; (**B**) injection method of tamoxifen administration. (**C**) Detection of the tamoxifen-induced *Cdk8* knockout in the mouse brain and uterus (original oral and improved injection methods of tamoxifen administration).

**Figure 2 ijms-23-14077-f002:**
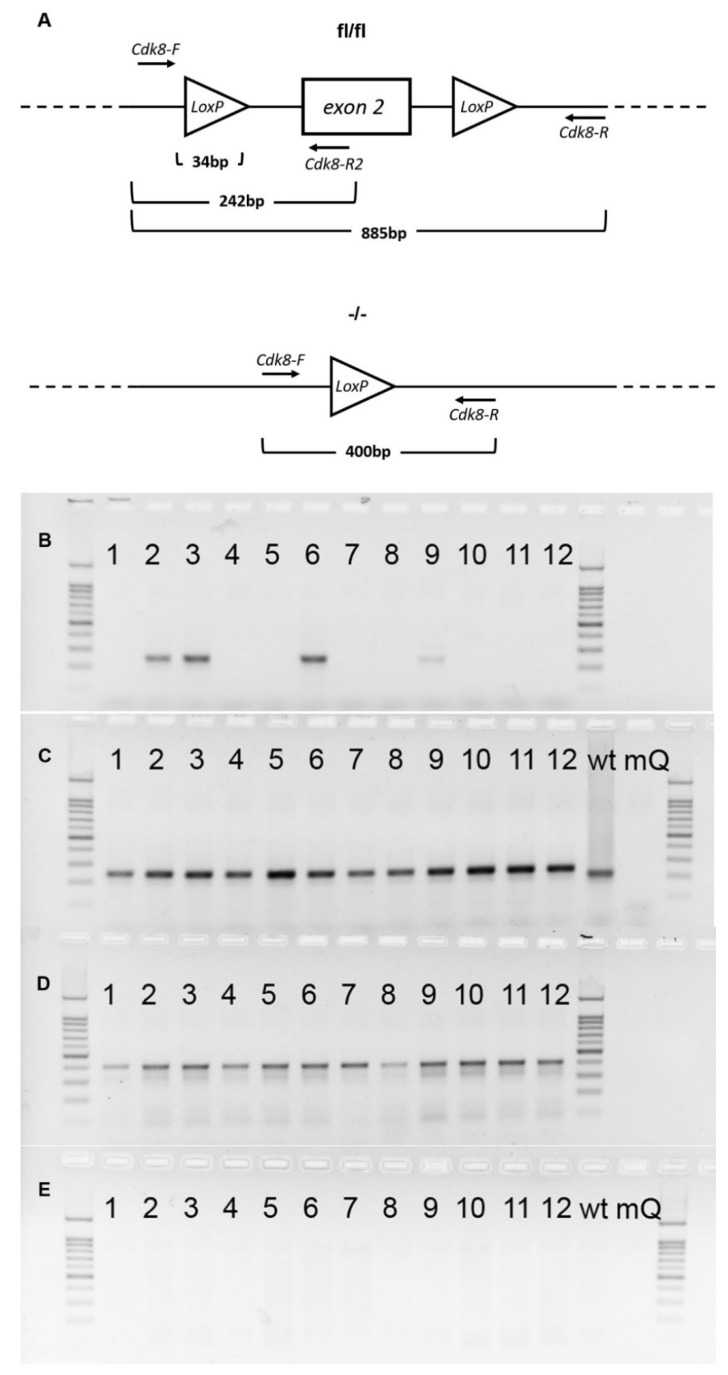
(**A**)—Schemes of loxP-sites localization in *Cdk8* gene of Cdk8^fl/fl^ mice (upper scheme) and Cdk8^−/−^ mice (lower scheme) and annealing positions of PCR primers used for genotyping of animals. (**D**,**E**)—PCR detection of 2nd *Cdk8* exon excision. Panels (**B**,**C**) represent CDK8-F + R2 PCR product, which is 242 bp (**A**) in case of floxed exon 2 presence and is absent when the exon is completely excised. (**D**,**E**)—from PCR with F + R primer pair that flanks the whole floxed exon (~890 bp). Panels (**B**,**D**) correspond to a set of genomic PCRs of organs from tamoxifen-treated Cdk8^fl/fl^ female mice. (**C**,**E**)—those of Cdk8^fl/fl^ control mouse. Organs are shown in the following order: 1—heart, 2—ovary, 3—uterus, 4—liver, 5—lung, 6—brain, 7—muscle (quadriceps), 8—kidney, 9—small intestine, 10—thymus, 11—spleen, 12—bone marrow. “wt”—wildtype mouse tail sample; “mQ”—no template control.

**Figure 3 ijms-23-14077-f003:**
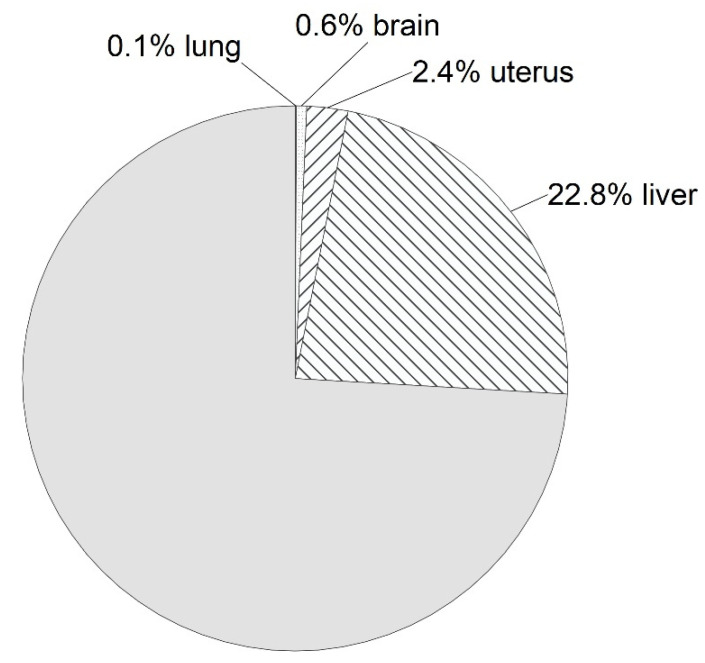
The 3 mg tamoxifen dose distribution between organs with respect to their weights at 4 h after injection.

**Figure 4 ijms-23-14077-f004:**
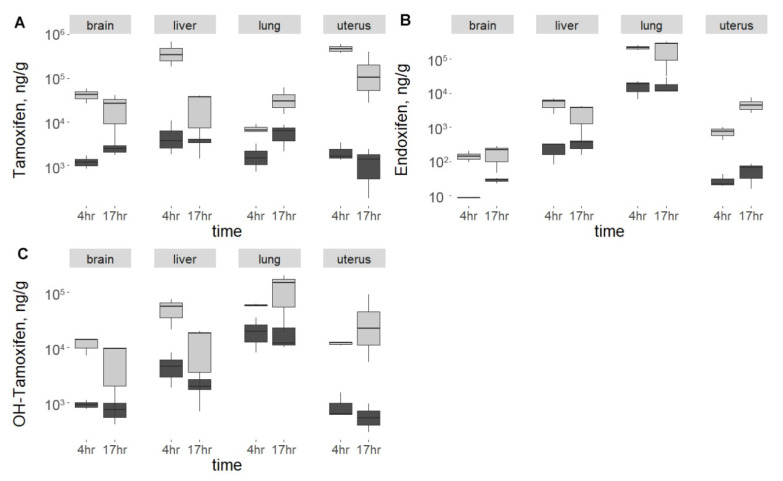
Tamoxifen (**A**) and its metabolites (**B**,**C**) concentrations in organs in exponential scale. Pale (□) boxes represent injection method, while feeding method samples are shown dark (■).

**Figure 5 ijms-23-14077-f005:**
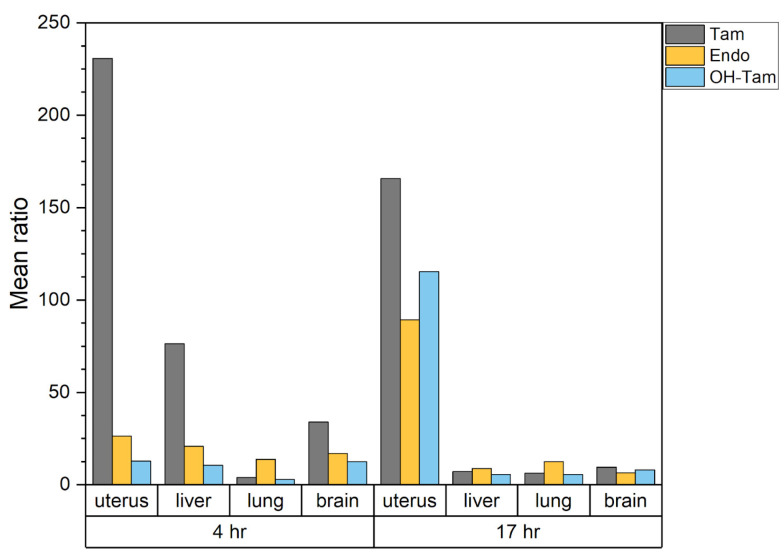
Ratios of mean organ metabolite concentration in injected and fed sample groups represented as columns.

**Figure 6 ijms-23-14077-f006:**
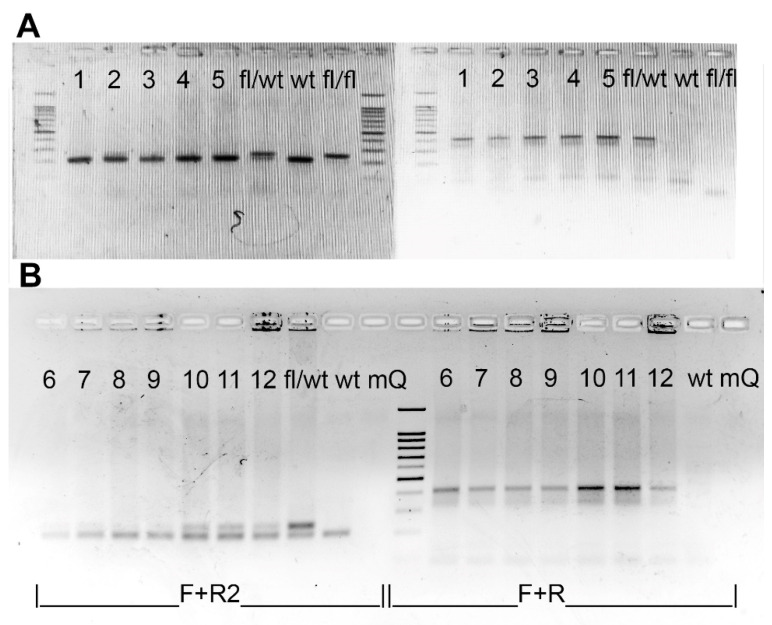
Analysis of exon 2 excision in embryos by DNA electrophoresis. Single 208 bp band without 242 bp band (left sides, “F + R2”) confirmed the presence of wild type allele and complete excision of floxed sequence, while presence of both bands indicates incomplete excision. 414 bp (right sides, “F + R”) confirms presence of edited allele. (**A**) Complete excision of exon 2 after tamoxifen administration starting at 10.5 dpc (lanes 1–5). (**B**) Incomplete excision (lanes 10–12) of exon 2 after tamoxifen administration starting at 17.5 dpc. “fl/wt”—treated heterozygous for a floxed exon mouse sample, “wt”—wildtype mouse, “fl/fl”—untreated homozygous for floxed exon mouse sample, mQ—no template control (milliQ water).

**Figure 7 ijms-23-14077-f007:**
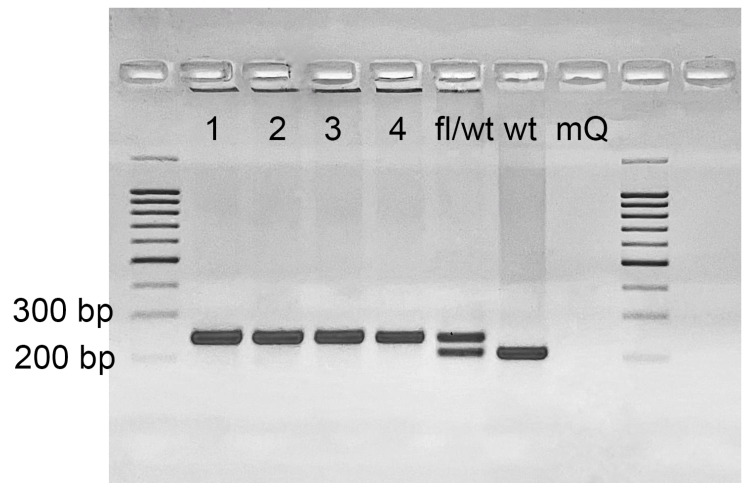
Genotyping of Cdk8^fl/fl^ mice. 208 bp band corresponds to the wild type allele and 242 band to the floxed allele. “fl/wt”—untreated heterozygous for a floxed exon mouse sample. “wt”—wildtype mouse. mQ—no template control (milliQ water). Lanes 1–4 with 242 bp bands give an example of homozygous for floxed exon samples.

**Table 1 ijms-23-14077-t001:** Ratios of mean organ metabolite concentration in injected and fed sample groups.

		Ratio of Mean Concentrations, Injection/Feeding		
Time, Hours	Organ	Tamoxifen	Endoxifen	OH-Tamoxifen
4	uterus	230.79	26.36	12.87
4	liver	76.26	20.81	10.58
4	lung	3.97	13.76	2.83
4	brain	33.96	16.96	12.41
17	uterus	165.82	89.33	115.36
17	liver	7.23	8.83	5.50
17	lung	6.30	12.42	5.59
17	brain	9.51	6.45	8.16
Mean		66.73	24.37	21.66
Median		21.76	15.36	9.37

**Table 2 ijms-23-14077-t002:** Effects of tamoxifen action on pregnancy and embryonic development in mice.

Mouse # and Genotype	Tamoxifen Dosage, Type of Administration	Stage of Pregnancy When Mice Started to Received Tamoxifen	Results
Mouse #1 F_1_ hybrid CBA × C57Bl/6J	2 days of 3 mg tamoxifen per day by oral administration	15.5 dpc	Uterine bleeding, premature birth, death of pups and mother.
Mouse #2 F_1_ hybrid CBA × C57Bl/6J	5 days of 1.5 mg tamoxifen per day by oral administration	15.5 dpc	Pathological labor, pups stuck in the birth canal, death of pups and mother.
Mouse #3 F_1_ hybrid CBA × C57Bl/6J	5 days of 1.5 mg tamoxifen per day by oral administration	10.5 dpc	Premature labor at 17.5 dpc, pups’ death.
Mice #4–6 C57Bl/6J	3 days of 1.5 mg tamoxifen per day by intraperitoneal infections	10.5dpc	Premature labor at 18.5 dpc, birth of dead and nonviable pups
Mice #7,8 C57Bl/6J	3 days of 1.5 mg tamoxifen per day by intraperitoneal infections	15.5 dpc	Premature labor at 18.5 dpc, birth of dead and nonviable pups

**Table 3 ijms-23-14077-t003:** HPLC/MS conditions for metabolite quantification in extracts.

	Collision Energy (CE), eV	Retention Time, Min	Mass/Charge (*m*/*z*)
Tamoxifen	24.0	3.16	371.9 > 72.1
Endoxifen	22.0	1.83	374.2 > 58.0
4-OH-tamoxifen	25.0	1.90	388.2 > 72.0

## Data Availability

Not applicable.

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
