# Peer review of "Limitations of Tamoxifen Application for In Vivo Genome Editing Using Cre/ERT2 System"

_ijms, 2022, doi:10.3390/ijms232214077_

Round 1

Reviewer 1 Report

The authors have studied the reaches of tamoxifen application in Cre/ERt2 system for in vivo editing regarding organs such as the lung, liver, spleen, and brain as well, they investigated the tamoxifen/Cre/ERt2 activity on embryo development. It is useful report, bringing more details for the strategic use of this genome editing tool in vivo. Although they confirmed some results already published about tamoxifen's teratogenic effect, several issues were reached that were very relevant, about different levels of effectiveness of the system in different organs, comparing methods of administration. An inquiring issue related to uterus effectiveness editing and concentration of tamoxifen was unexpected data that would need further investigation to better understanding. In general, the work is clear, with good results and discussion, and useful.

Author Response

Dear Ms. Abby Jiang,

We would like to thank the editor and the reviewers for the fast and thorough evaluation of our work, and for giving us the opportunity to resubmit the revised manuscript “Limitations of tamoxifen application for in vivo genome editing using Cre/ERT2 system”. We are grateful for the suggestions made by the reviewers and have revised the manuscript accordingly.

Here are the point-by-point replies to reviews.

Reviewer 1.

We would like to thank Reviewer 1 for the high evaluation of our paper. Regarding the comment Although they confirmed some results already published about tamoxifen's teratogenic effect.” we do agree that these issues were previously addressed in the literature, nevertheless, recent articles, such as “Yoshinobu et al., 2021” (reference [22] in our article) successfully used the tamoxifen method for in utero Cre/Lox recombination.

Also all modifications in the current version of the text of the article in comparison to the previous version are highlighted by red.

We again would like to thank the reviewers for the thorough examination of our article and the editorial team for the opportunity to submit our revised manuscript to IJMS. We look forward to addressing any additional comments made by the reviewers and revising the article in its final form.

Sincerely,

Filatov Maxim

Reviewer 2 Report

Ilchuk Leonid A. et al have submitted a comprehensive article focused on the Tamoxifen (TX) inducible system for producing tissue-specific conditional mouse knock out models. In their study they compared oral gavage versus intraperitoneal (IP) injection of tamoxifen as a method of administration. They analysed the knockout effectiveness using the mouse model Cdk8 floxed/floxed/Rosa-Cre-ERT2 mic and they also studied the tamoxifen tissue distribution. They noted less effectiveness for knocking out Cdk8 in brain and uterus with both methods, but increased doses of TX by IP administration achieved a total Cdk8 knock-out. In contrast, increased doses of TX by oral gavage did not get the same result. To clarify the causes of insufficient Cdk8 knockout activation in the brain and uterus they measured concentrations of tamoxifen and its main active metabolites. They discovered that oral gavage administration reaches lower metabolite concentrations than IP for each organ. They also described alterations on pregnancy that many other have described before in several papers.

Major questions: To increase the interest of this study to be attractive for the scientific community my suggestion is:

1)      To add a new figure showed the Cdk8 targeting locus (with wt and floxed alleles, and distance between loxP sites)

2)      To perform a set of genomic PCRs from several organs (heart, ovary, intestine, stomach, liver, spleen, lung, thymus, bone marrow, muscle, …) and show it in a new figure

3)      To measure the distribution of TX and active metabolites in all these organs and modify the figure 2 according new data.

Minor questions:

Line 103 “(see Fig. 1A and 2B)” should be replaced by (see Fig. 1A and 1B)

Author Response

Dear Ms. Abby Jiang,

We would like to thank the editor and the reviewers for the fast and thorough evaluation of our work, and for giving us the opportunity to resubmit the revised manuscript “Limitations of tamoxifen application for in vivo genome editing using Cre/ERT2 system”. We are grateful for the suggestions made by the reviewers and have revised the manuscript accordingly.

Here are the point-by-point replies to reviews.

Reviewer 2.

We would like to thank reviewer 2 for thorough evaluation of our paper. We thank the reviewer for pointing out several areas where the article could be improved.

1)         To add a new figure showed the Cdk8 targeting locus (with wt and floxed alleles, and distance between loxP sites).

We made a new figure 2A, making all suggested improvements.

2)         To perform a set of genomic PCRs from several organs (heart, ovary, intestine, stomach, liver, spleen, lung, thymus, bone marrow, muscle, …) and show it in a new figure.

We thank the reviewer for the important point raised. We performed genomic PCRs in multiple organs showing complete excision in all organs except brain, uterus, ovaries and intestine (see Fig 2B-E). We should note that PCR is a very sensitive method and detects unrestricted DNA even in absence of detectable protein (like in the brain and intestine). The text of the abstract and discussion was appropriately changed to reflect the new figure (pages 6, 12 and 13)

3)         To measure the distribution of TX and active metabolites in all these organs and modify the figure 2 according new data.

The reviewer raises an important issue, concerning the distribution of TX and its metabolites. Nevertheless the scope of our paper was to examine the organs where the excision was not complete and to improve, if possible, the rate of the knockout. Our data shows complete excision of flox sites in other organs and we do not consider that further examination of TX metabolism in them will provide new insights.

Minor questions:

Line 103 “(see Fig. 1A and 2B)” should be replaced by (see Fig. 1A and 1B)

Appropriate changes were made, according to the new figure order. Also all modifications in the current version of the text of the article in comparison to the previous version are highlighted by red.

We again would like to thank the reviewers for the thorough examination of our article and the editorial team for the opportunity to submit our revised manuscript to IJMS. We look forward to addressing any additional comments made by the reviewers and revising the article in its final form.

Sincerely,

Filatov Maxim